# Task Agnostic and Task Specific Self-Supervised Learning from Speech with *LeBenchmark*

**Solène Evain**[1,*], **Ha Nguyen**[1,2,*], **Hang Le**[1,*], **Marcely Zanon Boito**[1,2,*], **Salima Mdhaffar**[2,*], **Sina Alisamir**[1,3,*], **Ziyi Tong**[1], **Natalia Tomashenko**[2,*], **Marco Dinarelli**[1,*], **Titouan Parcollet**[2,*], **Alexandre Allauzen**[4], **Yannick Estève**[2], **Benjamin Lecouteux**[1], **François Portet**[1], **Solange Rossato**[1], **Fabien Ringeval**[1], **Didier Schwab**[1], and **Laurent Besacier**[5]

[1]Univ. Grenoble Alpes, CNRS, Inria, Grenoble INP, LIG, 38000 Grenoble, France
[2]LIA, Avignon Université, France
[3]Atos, Échirolles, France
[4]ESPCI, CNRS LAMSADE, PSL Research University, France
[5]Naver Labs Europe, France
[*]Equal contributors

## Abstract

Self-Supervised Learning (SSL) has yielded remarkable improvements in many different domains including computer vision, natural language processing and speech processing by leveraging large amounts of unlabeled data. In the specific context of speech, however, and despite promising results, there exists a clear lack of standardization in the evaluation process for comprehensive comparisons of these models. This issue gets even worse with the investigation of SSL approaches for other languages than English. We present *LeBenchmark*, an open-source and reproducible framework for assessing SSL from French speech data. It includes documented, large-scale and heterogeneous corpora, seven pretrained SSL wav2vec 2.0 models shared with the community, and a clear evaluation protocol made of four downstream tasks along with their scoring scripts: automatic speech recognition, spoken language understanding, automatic speech translation and automatic emotion recognition. For the first time, SSL models are analyzed and compared on the latter domains both from a task-agnostic (*i.e.* frozen) and task-specific (*i.e.* fine-tuned w.r.t the downstream task) perspectives. We report state-of-the-art performance on most considered French tasks and provide a readable evaluation set-up for the development of future SSL models for speech processing.

## 1 Introduction

Self-Supervised Learning (SSL) based on huge amounts of unlabeled data has been explored successfully for image and natural language processing [1, 2, 3, 4]. Recently, researchers investigated SSL from speech as well and successfully improved performance on downstream tasks such as speech recognition [5, 6]. As SSL from speech is a rapidly evolving domain, new models are unfortunately evaluated on different datasets, most of which focus on the English language. In order to carefully assess the progress of speech SSL model-wise and application-wise, common benchmarks are needed. While NLP benchmarking is now widely discussed [7], multi-task benchmarks are less common in speech despite the fact that the field has a long tradition of evaluation (see for instance long-term NIST and DARPA shared tasks for ASR). We propose to contribute to this by providing a reproducible and multifaceted benchmark for evaluating speech SSL models. By *benchmark*, and following the definition of [8], we mean an ensemble of tasks that allow to discriminate learners (*i.e.* SSL models) based on their ability to perform well on those tasks. We propose an initial set of four main tasks (10 sub-tasks overall), measuring specific speech challenges in French language: Automatic Speech

Recognition (ASR), Spoken Language Understanding (SLU), Speech Translation (AST) and Emotion Recognition (AER). This enables to assess the impact of pre-trained speech models that differ along several dimensions: language used for pre-training (French, English, multilingual), amount of raw speech used for SSL pre-training (1k, 3k or 7k hours), model size (base, large). For reproducibility, we also provide pre-trained SSL models learned on a large and heterogeneous collection of speech utterances and believe this is a strong contribution to speech technologies in French. This work extends a preliminary proposal [9] with a bigger speech corpus for SSL, more SSL models evaluated and shared, as well as experiments comparing task agnostic models (*i.e.* SSL models trained with pre-training objective on general purpose data) and task specific models (*i.e.* SSL models obtained after task-adaptive pre-training [10] or after fine-tuning for an ASR task). Our website shares models, scripts and results for better transparency and reproducibility of research in speech SSL.[1]

## 2 Background

SSL has been recently proposed as an interesting alternative for data representation learning, as it requires no annotated data. Such learned representations have been very successful in vision [1, 2] and NLP [3, 11]. SSL from speech consists of resolving *pseudo-tasks*, which do not require human annotation, as a pre-training for the real tasks to solve. These *pseudo-tasks* target predicting the next samples, or solving ordering problems. For instance, Autoregressive Predictive Coding (APC) considers the sequential structure of speech and predicts information about a future frame [12, 13], whereas Contrastive Predictive Coding (CPC) distinguishes a future speech frame from distractor samples [5, 14], which is an easier learning objective compared to APC. Such representations have been shown to improve performance in several speech tasks [15], while being less sensitive to domain and/or language mismatch [6] and being transferable to other languages [16]. In 2020, a strong speech SSL baseline appeared: the Wav2Vec2.0 model [17] which relies on the CPC idea of [5, 14] but with *discrete* speech units that are used as latent representations and fed to a Transformer network to build contextualized representations. Several other bi-directional encoders were also proposed recently: Speech-XLNet [18], Mockingjay [19] and [20]. A few recent studies were also related to multilingual SSL models trained on very large multilingual corpora [21, 22].

While there are multiple evaluation benchmarks to assess pretrained models in NLP (see for instance [23] for English, [24] for French and [25] for Korean), we are aware of only two similar initiatives for speech SSL models' evaluation: our own preliminary work [9] and the Speech processing Universal PERformance Benchmark (SUPERB) [26] which however targets English language only and does not share pre-trained SSL models as we do.

## 3 Gathering a Large and Heterogeneous Speech Collection in French

Recently, large multilingual corpora that include French have been made available, such as MLS [27] (1,096 h), or voxpopuli [22] (+4,500 h). However, these are restricted to either read or well-prepared speech, failing to provide diversity in the speech samples, such as accented, spontaneous and/or affective speech. In this work, we gathered a large variety of speech corpora in French that cover different accents (MLS, African Accented Speech, CaFE), acted emotions (GEMEP, CaFE, Att-Hack), telephone dialogues (PORTMEDIA), read (MLS, African Accented French, MaSS) and spontaneous sentences (CFPP2000, ESLO2, MPF, TCOF, NCCFr), broadcast speech (EPAC) and professional speech (Voxpopuli). Compared to MLS and Voxpopuli, our dataset is more diverse, carefully sourced and contains detailed metadata (speech type, and speaker gender). Moreover, it has a more realistic representation of speech turns in real life, compared to MLS and VoxPopuli (see average utterance duration in Table 1). Statistics are reported in Table 1.

**Pre-processing for SSL training:** Recordings were segmented using time stamps from transcriptions. We retrieved, when available, speaker labels and gender information. Following [17], we removed utterances shorter than 1 s, and longer than 30 s. When possible, overlapping speech sentences were also removed. When necessary, audio segments were converted to mono PCM 16 bits, 16 kHz.

**Small dataset ($\approx$ 1k hours)** is only composed of the MLS corpus for comparison with Wav2Vec2.0 [17] which uses only read English speech. It is also gender balanced.

---

[1] http://lebenchmark.com

Table 1: Statistics for the speech corpora used to train SSL models according to gender information (male / female / unknown). The small dataset is from MLS only. Every dataset is composed of the previous one + additional data; duration: hour(s):minute(s).

| Corpus$_{License}$ | # Utterances | Duration | # Speakers | Mean Utt. Duration | Speech type |
|---|---|---|---|---|---|
| | | **Small dataset – 1K** | | | |
| MLS French$_{CCBY4.0}$ [27] | **263,055** 
 124,590 / 138,465 / – | **1,096:43** 
 520:13 / 576:29 / – | **178** 
 80 / 98 / – | **15 s** 
 15 s / 15 s / – | Read |
| | | **Medium dataset – 3K** | | | |
| African Accented French$_{Apache2.0}$ [28] | **16,402** 
 373 / 102 / 15,927 | **18:56** 
 – / – / 18:56 | **232** 
 48 / 36 / 148 | **4 s** 
 – / – / – | Read |
| Att-Hack$_{CCBYNCND}$ [29] | **36,339** 
 16,564 / 19,775 / – | **27:02** 
 12:07 / 14:54 / – | **20** 
 9 / 11 / – | **2.7 s** 
 2.6 s / 2.7 s / – | Acted Emotional |
| CaFE$_{CCNC}$ [30] | **936** 
 468 / 468 / – | **1:09** 
 0:32 / 0:36 / – | **12** 
 6 / 6 / – | **4.4 s** 
 4.2 s / 4.7 s / – | Acted Emotional |
| CFPP2000$_{CCBYNCSA}$* [31] | **9853** 
 166 / 1,184 / 8,503 | **16:26** 
 0:14 / 1:56 / 14:16 | **49** 
 2 / 4 / 43 | **6 s** 
 5 s / 5 s / 6 s | Spontaneous |
| ESLO2$_{NC}$ [32] | **62,918** 
 30,440 / 32,147 / 331 | **34:12** 
 17:06 / 16:57 / 0:09 | **190** 
 68 / 120 / 2 | **1.9 s** 
 2 s / 1.9 s / 1.7 s | Spontaneous |
| EPAC**$_{NC}$ [33] | **623,250** 
 465,859 / 157,391 / – | **1,626:02** 
 1,240:10 / 385:52 / – | **Unk** 
 – / – / – | **9 s** 
 – / – / – | Radio Broadcasts |
| GEMEP$_{NC}$ [34] | **1,236** 
 616 / 620 / – | **0:50** 
 0:24 / 0:26 / – | **10** 
 5 / 5 / – | **2.5 s** 
 2.4 s / 2.5 s / – | Acted Emotional |
| MPF [35], [36] | **19,527** 
 5,326 / 4,649 / 9,552 | **19:06** 
 5:26 / 4:36 / 9:03 | **114** 
 36 / 29 / 49 | **3.5 s** 
 3.7 s / 3.6 s / 3.4 s | Spontaneous |
| PORTMEDIA$_{NC}$ (French) [37] | **19,627** 
 9,294 / 10,333 / – | **38:59** 
 19:08 / 19:50 / – | **193** 
 84 / 109 / – | **7.1 s** 
 7.4 s / 6.9 s / – | Acted telephone dialogue |
| TCOF (Adults) [38] | **58,722** 
 10,377 / 14,763 / 33,582 | **53:59** 
 9:33 / 12:39 / 31:46 | **749** 
 119 / 162 / 468 | **3.3 s** 
 3.3 s / 3.1 s / 3.4 s | Spontaneous |
| **Medium dataset total** | **1,111,865** 
 664,073 / 379,897 / 67,895 | **2,933:24** 
 1,824:53 / 1,034:15 / 74:10 | - | - | - |
| | | **Large dataset – 7K** | | | |
| MaSS [39] | **8,219** 
 8,219 / – / – | **19:40** 
 19:40 / – / – | **Unk** 
 – / – / – | **8.6 s** 
 8.6 s / – / – | Read |
| NCCFr$_{NC}$ [40] | **29,421** 
 14,570 / 13,922 / 929 | **26:35** 
 12:44 / 12:59 / 00:50 | **46** 
 24 / 21 / 1 | **3 s** 
 3 s / 3 s / 3 s | Spontaneous |
| Voxpopuli$_{CC0}$ [22] *Unlabeled* | **568,338** 
 – / – / – | **4,532:17** 
 – / – / 4,532:17 | **Unk** 
 – / – / – | **29 s** 
 – / – / – | Professional speech |
| Voxpopuli$_{CC0}$ [22] *transcribed* | **76,281** 
 – / – / – | **211:57** 
 – / – / 211:57 | **327** 
 – / – / – | **10 s** 
 – / – / – | Professional speech |
| **Large dataset total*** | **1,814,242** 
 682,322 / 388,217 / 99,084 | **7,739:22** 
 1,853:02 / 1,041:07 / 4,845:07 | - | - | - |

*Composed of audio files not included in the CEFC corpus v2.1, 02/2021; **speakers are not uniquely identified.; ***Stats of CFPP2000, MPF and TCOF have changed a bit due to a change in data extraction; License: CC=Creative Commons; NC=non-commercial; BY= Attribution; SA= Share Alike; ND = No Derivative works; CC0 = No Rights Reserved

**Medium dataset (≈ 3k hours)** includes 2,933 h of speech, from which 1,115 h is read speech, 1,626 h broadcast speech, 123 h spontaneous speech, 38 h acted telephone dialogues, and 29 h acted emotional speech. Regarding gender, we collected 1,824 h of speech from male speakers, 1,034 h from female speakers, and 74 h from unknown gender.

**Large dataset (≈ 7.7k hours)** has 4 additional corpora: MaSS, NCCFr and Voxpopuli (unlabeled + transcribed). It includes 7,739 h of speech, from which 1,135 h is read speech, 1,626 h broadcast speech, 165 h spontaneous speech, 38 h acted telephone dialogues, 29 h acted emotional speech, and 4744 h professional speech. Except for NCCFr, no info about gender is given in the added datasets.

## 4 Training and Sharing SSL Models

*LeBenchmark* provides seven Wav2Vec2.0 models [17] pretrained on the gathered French data described in Section 3. Following [17], two different Wav2Vec2.0 architectures (*large* and *base*) are coupled with our *small* (1K), *medium* (3K) and *large* (7K) corpus to form our set of Wav2Vec2.0 models: W2V2-Fr-1K-*base*, W2V2-Fr-1K-*large*, W2V2-Fr-3K-*base*, W2V2-Fr-3K-*large*, W2V2-Fr-7K-*base*, W2V2-Fr-7K-*large*. We also provide a specific model (W2V2-Fr-2.7K-*base*) trained on a subset of our *medium* set only containing MLS and EPAC (2.7K hours of audio) to enable further investigation on the impact of spontaneous speech on SSL representations.

Hyperparameters and architectures for *base*[2] and *large*[3] are identical to the ones first introduced in [17]. W2V2-Fr-1K, W2V2-Fr-3K, W2V2-Fr-2.7K and W2V2-Fr-7K are trained respectively for 200K, 500K, 500K and 500K updates on 4, 32, 32 and 64 Nvidia Tesla V100 (32GB), with one

---

[2] https://github.com/pytorch/fairseq/blob/main/examples/wav2vec/config/ pretraining/wav2vec2_base_librispeech.yaml

[3] https://github.com/pytorch/fairseq/blob/main/examples/wav2vec/config/ pretraining/wav2vec2_large_librivox.yaml

update corresponding to a call to the *.backward()* function in PyTorch. Detailed summary of the hyperparameters used to train our SSL models can be found in Table 2. In practice, training is stopped at a round number of updates once the loss observed on the development set of the MLS corpus reaches a stable point (learning curves are given in Appendix A.1).

Pre-trained Wav2Vec2.0 models are shared with the community via HuggingFace[4] for further integration with well-known toolkits such as SpeechBrain [41], Fairseq [42] or Kaldi [43].

Pre-existing Wav2Vec2.0 models obtained from Fairseq[5] are also considered in downstream experiments. First, XLSR-53-*large* is used as a comparison to multilingual models. Then, W2V2-En-*base* and W2V2-En-*large* (LS960) are used to assess English representations from LibriSpeech.[6]

Table 2: Hyperparameters of our pre-trained SSL models

| Model | Training data | Transformer blocks | Model dimension | Inner dimension | Heads | Updates |
|---|---|---|---|---|---|---|
| W2V2-Fr-1K-*base* | 1,096 h | 12 | 768 | 3,072 | 8 | 200K |
| W2V2-Fr-1K-*large* | 1,096 h | 24 | 1024 | 4,096 | 16 | 200K |
| W2V2-Fr-2.7K-*base* | 2,773 h | 12 | 768 | 3,072 | 8 | 500K |
| W2V2-Fr-3K-*base* | 2,933 h | 12 | 768 | 3,072 | 8 | 500K |
| W2V2-Fr-3K-*large* | 2,933 h | 24 | 1024 | 4,096 | 16 | 500K |
| W2V2-Fr-7K-*base* | 7,739 h | 12 | 768 | 3,072 | 8 | 500K |
| W2V2-Fr-7K-*large* | 7,739 h | 24 | 1,024 | 4,096 | 16 | 500K |

## 5 Benchmarking SSL Models

We benchmark SSL models on four different tasks (ASR, SLU, AST and AER) chosen with respect to following criteria: (a) diversity of problems: regression (AER), sequence labelling (SLU) and conditional natural language generation (ASR, AST), (b) diversity of information extracted: transcript (ASR), semantics (SLU), translation (AST) and paralinguistics (AER), and (c) diversity of annotated resources available for downstream task: large (ASR), medium (SLU, AST), small (AER). As our goal is to evaluate the impact of SSL for the best baselines for each task addressed, we have a different architecture for each task and it corresponds to the best baseline performance we could obtain using MFCC/FBANK features. As a different architecture/approach is used for each task, we evaluate the different SSL models as feature extractors for these tasks. These 'SSL extractors' are either 'task agnostic' or 'task specific' (SSL models fine-tuned on the task data) as further explained below.

### 5.1 Automatic Speech Recognition (ASR)

SSL for ASR is evaluated using both hybrid DNN-HMM and end-to-end approaches. In addition to the source code used to make these ASR experiments (training + decoding), *LeBenchmark* provides a normalization script for French output text derived from the one applied during the official French ESTER and ETAPE evaluation campaigns [44] and a unique script to compute the Word Error Rate (WER) from ASR output.

**Datasets** The ASR tasks target two different types of corpora: Common Voice [45] and ETAPE [44]. Common Voice is a very large crowd-sourced corpus (477 h) of read speech in French with transcripts – train: 428 h, dev: 24 h, and test: 25 h, while ETAPE is a smaller (36 h) but more challenging corpus composed of diverse French TV broadcast programs – train: 22 h, dev: 7 h, and test: 7 h.

**Hybrid DNN-HMM** The acoustic models (AM) are trained on 40-dimensional high-resolution (*hires*) MFCC features or SSL features using the Kaldi toolkit [43] with a state-of-the-art factorized time delay neural network (TDNN-F) architecture [46, 47] on the ETAPE training corpus [44] only. More details about the AM architecture are given in Appendix A.2.1. Two trigram LMs were used in

---

[4]https://huggingface.co/LeBenchmark
[5]https://github.com/pytorch/fairseq/tree/master/examples/wav2vec
[6]For the sake of conciseness, we remove the prefix *W2V2-* in all our results table.

Table 3: ASR results (WER,%) on the ETAPE corpus for hybrid DNN-HMM AM with TDNN-F topology. Gray numbers indicate 95% confidence intervals.[8]

| Language Model | ETAPE | | ESTER-1.2 + EPAC | |
|---|---|---|---|---|
| **Features** | **Dev** | **Test** | **Dev** | **Test** |
| hires MFCC | 36.89±0.66 | 38.50±0.71 | 29.56±0.70 | 31.93±0.75 |
| **(a) Task-agnostic pre-training** | | | | |
| En-*large* | 37.68±0.71 | 40.31±0.75 | 30.51±0.73 | 33.32±0.79 |
| XLSR-53-*large* | 34.28±0.69 | 36.03±0.72 | 27.01±0.68 | 29.64±0.77 |
| Fr-1K-*base* | 38.91±0.72 | 41.53±0.80 | 32.26±0.74 | 35.69±0.82 |
| Fr-1K-*large* | 38.77±0.71 | 40.69±0.67 | 32.29±0.73 | 34.91±0.79 |
| Fr-2.7K-*base* | 32.35±0.66 | 34.43±0.72 | 26.65±0.67 | 29.31±0.74 |
| Fr-3K-*base* | 31.98±0.66 | 33.61±0.73 | 25.83±0.66 | 27.82±0.74 |
| Fr-3K-*large* | 31.85±0.64 | 33.46±0.69 | 26.54±0.65 | 28.56±0.72 |
| Fr-7K-*base* | 31.96±0.67 | 33.36±0.72 | 26.03±0.67 | 27.09±0.76 |
| Fr-7K-*large* | **28.75**±0.62 | **30.30**±0.68 | **23.62**±0.63 | **25.64**±0.70 |
| **(c) Task-specific pre-training (fine-tuned for ASR on ETAPE)** | | | | |
| Fr-2.7K-*base* | 32.34±0.64 | 34.46±0.73 | 26.44±0.66 | 29.11±0.75 |
| Fr-3K-*base* | 31.89±0.64 | 33.47±0.71 | 26.12±0.66 | 28.03±0.75 |
| Fr-3K-*large* | **28.82**±0.62 | **30.19**±0.67 | 23.67±0.62 | **25.22**±0.70 |
| Fr-7K-*base* | 31.70±0.65 | 33.32±0.73 | 25.84±0.67 | 28.24±0.76 |
| Fr-7K-*large* | 28.84±0.61 | 30.29±0.66 | **23.44**±0.62 | 25.36±0.70 |

evaluation: (1) trained on ESTER-1.2 and EPAC training data (with a 82k vocabulary) and (2) trained on ETAPE training data only (with a smaller 17.5k vocabulary).

**End-to-End** Our end-to-end (e2e) systems are implemented with SpeechBrain toolkit [41]. The baseline e2e system is fed by 80-dimension log Mel filterbank (MFB) features and based on an encoder/decoder architecture with attention. When used with a SSL pre-trained Wav2Vec2.0 model, the e2e system simply adds an additional hidden layer and an output layer on top of Wav2Vec2.0 architecture. Details are given in Appendix A.2.2.

**Results** The WER results on the ETAPE development and test data sets for the hybrid DNN-HMM models are given in Table 3. Among the models trained on SSL features (Table 3, (a)) 6 models provide improvement over the baseline AM trained on MFCC features: XLSR-53, Fr-2.7k-*base*, Fr-3k-*base*, Fr-3k-*large*, Fr-7k-*base*, and Fr-7k-*large*. The best SSL features are the ones from the Fr-7k models and they clearly outperform the multilingual XLSR-53-*large*. In the case of task-specific pre-training,[7] we were not able to significantly improve the best results compared to task-agnostic pre-training. This is probably due to the fact that the obtained representations are not very different in both cases. These results can be compared to the ones obtained by the best ASR system during the official ETAPE shared task: by using 511h of training data (external training data were allowed), their ASR system got a word error rate of 23.6% [48], while in the experiments presented in this paper, only 22h of ETAPE training data were used. In the next paragraph, we investigate e2e fine-tuning of the models using transcribed speech.

Table 4 presents the results achieved with e2e ASR systems on French Common Voice 6.1 and on ETAPE. Before the use of Wav2vec2.0 models for ASR, the baseline MFB-based system (first line) was the state-of-the-art e2e model on CommonVoice/French. Other lines of table present different Wav2vec2.0 models fine-tuned on labeled ASR data from CommonVoice or ETAPE. Wav2vec2.0 *base* and *large* models provided by *LeBenchmark* outperform clearly En-*large* and XLSR-53-*large* models. The best model is Fr-3K-*large*, pretrained on a smaller training dataset than Fr-7K-*large*, and it provides the best results on all the experiments. We analyze gender performance in Appendix A.3 and show that female WER is systematically lower than male WER for all systems. Even for our Fr-3K SSL models trained with 38% of female speech only, female WER are particularly low.

## 5.2 Spoken Language Understanding (SLU)

**Dataset.** Spoken Language Understanding (SLU) aims at extracting a semantic representation from a speech signal in human-computer interaction applications [50, 51, 52, 53, 54]. Given the difficulty of

---

[7]Since two types of task-specific pre-training will be provided for SLU and AST, for ASR we only experimented with fine-tuning SSL models for ASR on ETAPE and then using them as feature extractors.

[8]Error margins corresponding to 95% confidence intervals were computed using bootstrap re-sampling as proposed in [49].

Table 4: End-to-end ASR results (WER%) on Common Voice and ETAPE corpora, with pre-trained wav2vec2.0 models further fine tuned on labeled ASR data.

| Corpus | CommonVoice | | ETAPE | |
|---|---|---|---|---|
| Features | Dev | Test | Dev | Test |
| MFB | 17.67±0.37 | 20.59±0.41 | 54.03±1.33 | 54.36±1.32 |
| En-*large* | 12.05±0.23 | 14.17±0.52 | 42.14±0.72 | 44.82±0.74 |
| XLSR-53-*large* | 16.41±0.27 | 19.40±0.29 | 58.55±0.65 | 61.03±0.70 |
| Fr-2.7K-*base* | 11.04±0.27 | 13.09±0.24 | 26.23±0.78 | 29.08±0.80 |
| Fr-3K-*base* | 11.25±0.23 | 13.22±0.24 | 26.14±0.70 | 28.86±0.79 |
| Fr-3K-*large* | **8.34**±0.18 | **9.75**±0.20 | **23.51**±0.68 | **26.14**±0.77 |
| Fr-7K-*base* | 10.84±0.21 | 12.88±0.24 | 25.13±0.68 | 28.16±0.79 |
| Fr-7K-*large* | 8.55±0.18 | 9.94±0.21 | 24.14±0.70 | 27.25±0.78 |

Table 5: End-to-end SLU decoding results (Concept Error Rate %) on the MEDIA corpus.

| Features | Dev | Test |
|---|---|---|
| (from [9]) spectrogram | 33.63±1.28 | 34.76±0.83 |
| spectrogram | **29.07**±1.31 | **31.10**±0.83 |
| **(a) Task agnostic pre-training** | | |
| En-*base* | 22.38±1.24 | 20.84±0.68 |
| En-*large* | 23.31±1.31 | 25.26±0.77 |
| Fr-1K-*base* | 22.89±1.26 | 23.27±0.76 |
| Fr-1K-*large* | 20.10±1.10 | 20.66±0.72 |
| Fr-2.7K-*base* | 18.63±1.13 | 18.42±0.65 |
| Fr-3K-*base* | 19.44±1.11 | 18.56±0.67 |
| Fr-3K-*large* | **15.96**±1.02 | **15.95**±0.62 |
| Fr-7K-*base* | 20.70±1.07 | 18.86±0.68 |
| Fr-7K-*large* | 17.25±1.02 | 16.35±0.66 |
| XLSR-53-*large* | 18.45±1.15 | 18.78±0.66 |
| **(b) Task specific pre-training (self-supervised on MEDIA)** | | |
| Fr-3K-*large* | 15.93±1.01 | **14.94**±0.60 |
| Fr-7K-*large* | **15.42**±1.03 | 15.17±0.60 |
| XLSR-53-*large* | 16.77±1.09 | 15.56±0.61 |
| **(c) Task specific pre-training (fine-tuned for ASR on MEDIA)** | | |
| Fr-3K-*large* | **14.49**±1.06 | 13.97±0.59 |
| Fr-7K-*large* | 14.58±1.01 | **13.78**±0.58 |
| XLSR-53-*large* | 16.05±1.05 | 15.46±0.60 |

creating an open-domain SLU application, many works focus on specific domains. We focus on the hotel information and reservation domain provided within the French corpus MEDIA [55, 56]. This corpus is made of 1 250 human-machine dialogues acquired with a *Wizard-of-Oz* approach, where 250 users followed 5 different reservation scenarios. Spoken data were manually transcribed and annotated with domain concepts, following a rich ontology. The official corpus split is made up of 12,908 utterances (41.5 h) for training, 1,259 utterances (3.5 h) for development and 3,005 utterances (11.3 h) for test. We note that, while all turns have been manually transcribed and can be used to train ASR models, only user turns have been annotated with concepts and can be used to train SLU models. This results in only 41.5 hours of speech training data for ASR models, and only 16.8 hours for SLU models.

**Experiments.** All our models are based on LSTM [57] seq2seq with attention [58]. Model details and training strategy are described in Appendix A.2.3. We use a total of 3 bidirectional LSTM layers of size 256 stacked in a pyramidal fashion in our encoder and the LSTM decoder has 2 layers of size 256. In addition to using spectrogram features and features from task agnostic SSL models, we also use features from task specific models (SLU on MEDIA). Two types of task-specific pre-training are performed: *self-supervised* which consists in resuming the SSL model training using the MEDIA training data and minimizing the *Wav2Vec 2.0* loss ('(b) self-supervised on MEDIA' in the table, also called task-adaptive pre-training in [10]); and *ASR supervised* ((c) fine-tuned for ASR on MEDIA in the table) which consists in fine-tuning the full SSL model for a supervised downstream task with a CTC loss minimization objective [59]. In this work we chose to fine-tune models with respect to the ASR task on MEDIA (not the SLU one) to see how it compares to self-supervised fine-tuning. We leave fine-tuning with respect to SLU for future work.

**Results** for SLU obtained with different speech representations are shown in Table 5. They are given in terms of Concept Error Rate (CER), computed the same way as Word Error Rate (WER) but on concept sequences. CER are accompanied by standard deviations (in gray), computed with the bootstrap method of [49]. We provide ASR results in supplementary material (table 10). We first note that our *spectrogram* baseline obtains a substantial improvement over the one in [9]. Such gain is due to the slightly different settings and model architecture described in the Appendix. Using SSL model features as input resulted in an impressive drop in CER, even when using English SSL models (CER from 31.10 to 20.84 on the test set with the *base* model). At best, among task-agnostic pre-trained models, we achieve a CER of 15.95 on the test data with Fr-3K-*large* features. Surprisingly, using features from the model trained with 7k hours of speech (Fr-7K-*large*), results are worse on both dev and test. In contrast, the 7k-model led to the best results in terms of ASR evaluation (see Table 10 in the Appendix). We performed task-specific pre-training only with the most effective SSL models: French 3k and 7k models and multi-lingual XLSR-53-*large*. The best overall pre-trained model is the 7k-model fine-tuned for ASR on MEDIA, though results are close to those obtained with features from the 3k-model (13.97 vs. 13.78). Indeed, significance tests in table 11 in the Appendix confirm that these two models are equivalent and they are significantly better than all the others. This shows that pre-trained SSL speech models can be specialized using task specific pre-training with either self-supervised learning on raw speech (block (b) in the table), or fine-tuning on raw speech and associated transcripts (block (c) in the table), the latter being slightly better than the former.

### 5.3 Automatic Speech-to-text Translation (AST)

Automatic speech-to-text translation (AST) consists in translating a speech utterance in a source language to a text in a target language. In this work, we are interested in translating directly from French speech to text in another language.

**Dataset** We selected subsets having French as the source in the multilingual TEDx dataset [60]. Our benchmark covers translation directions from French to three target languages: English (en), Spanish (es), and Portuguese (pt), with following training sizes 50 h (en), 38 h (es), and 25 h (pt).

**Experiments** Our baselines are models using 80-dimensional MFB features. For learned representations derived from SSL models, we focused on the feature extraction approach where features are extracted from either task-agnostic or task-specific pre-training. Task-agnostic pre-training refers to the direct use of SSL models as feature extractors whereas task-specific method consists in one additional phase where the SSL models are further trained on the in-domain task data, with (supervised fine-tuned) or without (self-supervised fine-tuned) labels. We performed supervised fine-tuning with speech transcriptions as labels and leave supervised fine-tuning with AST data for future work. In the task-specific scenario, we only considered three SSL models: two best French SSL models (Fr-3K-*large* and Fr-7K-*large*) and one best non-French SSL model (XLSR-53-*large*). Since the French speech is overlapped between the language pairs, we selected the pair having the most speech data (fr-en) to perform task-specific pre-training and used the obtained models to extract features for the remaining pairs (fr-es and fr-pt). For a fair comparison, we did not use additional data augmentation technique nor ASR encoder pre-training in the experiments. We refer to Appendix A.2.4 for details on the model architecture and implementation.

**Results** Table 6 displays the results of AST experiments. One can observe that SSL features, whether task-agnostic or task-specific and whether being pre-trained on English, French, or multilingual data, outperform the baselines using MFB features by a large margin (except for the task-agnostic multilingual model XLSR-53 on the two pairs fr-es and fr-pt, which are in very low-resource settings). Among the three groups using SSL features (task-agnostic pre-training, task-specific self-supervised, and task-specific fine-tuned for ASR), the ASR fine-tuning approach (c) yields the best results. We observe considerable improvements from task-specific self-supervised (b) to task-specific fine-tuned (c) (+6.19, +8.50, +8.53 on average for en, es, and pt, respectively) while the benefits of using self-supervised fine-tuning compared to task-agnostic pre-training are only marginal or even slightly negative. The substantial gains when using supervised fine-tuning approach (even with a somehow indirect signal which is transcripts for the AST downstream task) shows that giving more signals of the task-specific data to the SSL models is helpful. In particular, in the case of task-specific self-supervised fine-tuning (b), we further trained the SSL models for more steps on the raw task-specific data whereas in ASR fine-tuned scenario (c), we used raw data plus the transcripts to guide the SSL models. Focusing on task-agnostic block (a), we see that French SSL models

Table 6: BLEU on valid and test sets of multilingual TEDx (mTEDx). The highest value in each group (task-agnostic pre-training, task-specific self-supervised, and supervised fine-tuning) is underlined while the best value in each column is highlighted in **bold**. Gray numbers denote the standard deviation computed using bootstrap re-sampling [61].

| Features | Valid | | | Test | | |
|---|---|---|---|---|---|---|
| | **en** | **es** | **pt** | **en** | **es** | **pt** |
| MFB | $1.15_{\pm0.17}$ | $0.67_{\pm0.15}$ | $0.61_{\pm0.13}$ | $1.10_{\pm0.14}$ | $0.87_{\pm0.12}$ | $0.32_{\pm0.03}$ |
| **(a) Task agnostic pre-training** | | | | | | |
| En-*base* | $5.54_{\pm0.27}$ | $1.30_{\pm0.17}$ | $0.54_{\pm0.11}$ | $5.20_{\pm0.28}$ | $1.47_{\pm0.15}$ | $0.38_{\pm0.05}$ |
| En-*large* | $4.11_{\pm0.25}$ | $1.67_{\pm0.20}$ | $0.32_{\pm0.03}$ | $3.56_{\pm0.22}$ | $2.29_{\pm0.18}$ | $0.43_{\pm0.05}$ |
| Fr-1K-*base* | $9.18_{\pm0.36}$ | $5.09_{\pm0.27}$ | $0.39_{\pm0.05}$ | $8.98_{\pm0.36}$ | $5.64_{\pm0.30}$ | $0.49_{\pm0.08}$ |
| Fr-1K-*large* | $15.31_{\pm0.46}$ | $13.74_{\pm0.43}$ | $8.29_{\pm0.34}$ | $14.46_{\pm0.46}$ | $14.77_{\pm0.46}$ | $9.37_{\pm0.38}$ |
| Fr-2.7K-*base* | $15.09_{\pm0.49}$ | $13.27_{\pm0.43}$ | $4.72_{\pm0.27}$ | $14.69_{\pm0.48}$ | $14.04_{\pm0.43}$ | $5.51_{\pm0.28}$ |
| Fr-3K-*base* | $15.05_{\pm0.49}$ | $13.19_{\pm0.44}$ | $4.44_{\pm0.29}$ | $14.80_{\pm0.47}$ | $14.27_{\pm0.44}$ | $4.72_{\pm0.25}$ |
| Fr-3K-*large* | $17.94_{\pm0.51}$ | $16.40_{\pm0.49}$ | $8.64_{\pm0.34}$ | $18.00_{\pm0.51}$ | $18.12_{\pm0.48}$ | $9.55_{\pm0.36}$ |
| Fr-7K-*base* | $15.13_{\pm0.45}$ | $12.78_{\pm0.40}$ | $2.65_{\pm0.20}$ | $14.50_{\pm0.45}$ | $13.61_{\pm0.44}$ | $2.66_{\pm0.23}$ |
| Fr-7K-*large* | $\underline{19.23}_{\pm0.54}$ | $\underline{17.59}_{\pm0.49}$ | $\underline{9.68}_{\pm0.37}$ | $\underline{19.04}_{\pm0.53}$ | $\underline{18.24}_{\pm0.49}$ | $\underline{10.98}_{\pm0.41}$ |
| XLSR-53-*large* | $7.81_{\pm0.33}$ | $0.49_{\pm0.13}$ | $0.43_{\pm0.07}$ | $6.75_{\pm0.29}$ | $0.52_{\pm0.08}$ | $0.36_{\pm0.05}$ |
| **(b) Task specific pre-training (self-supervised on mTEDx)** | | | | | | |
| Fr-3K-*large* | $18.54_{\pm0.53}$ | $16.40_{\pm0.48}$ | $8.81_{\pm0.36}$ | $18.38_{\pm0.52}$ | $17.84_{\pm0.48}$ | $10.57_{\pm0.41}$ |
| Fr-7K-*large* | $\underline{19.65}_{\pm0.55}$ | $\underline{17.53}_{\pm0.47}$ | $\underline{9.35}_{\pm0.36}$ | $\underline{19.36}_{\pm0.54}$ | $\underline{18.95}_{\pm0.53}$ | $\underline{10.94}_{\pm0.38}$ |
| XLSR-53-*large* | $6.83_{\pm0.33}$ | $0.54_{\pm0.14}$ | $0.34_{\pm0.03}$ | $6.75_{\pm0.32}$ | $0.34_{\pm0.03}$ | $0.29_{\pm0.03}$ |
| **(c) Task specific pre-training (fine-tuned for ASR on mTEDx)** | | | | | | |
| Fr-3K-*large* | $21.09_{\pm0.53}$ | $19.28_{\pm0.53}$ | $14.40_{\pm0.47}$ | $21.34_{\pm0.58}$ | $21.18_{\pm0.52}$ | $16.66_{\pm0.49}$ |
| Fr-7K-*large* | $\mathbf{21.41}_{\pm0.51}$ | $20.32_{\pm0.49}$ | $\mathbf{15.14}_{\pm0.48}$ | $\mathbf{21.69}_{\pm0.58}$ | $\mathbf{21.57}_{\pm0.52}$ | $\mathbf{17.43}_{\pm0.52}$ |
| XLSR-53-*large* | $21.09_{\pm0.54}$ | $\mathbf{20.38}_{\pm0.56}$ | $14.56_{\pm0.45}$ | $20.68_{\pm0.53}$ | $21.14_{\pm0.55}$ | $17.21_{\pm0.54}$ |

clearly outperform those pre-trained on English and multilingual data. Multilingual XLSR-53 model surpasses the English models on fr-en, yet all of them fail to generate meaningful translations on fr-es and fr-pt where little training data is available. Comparing across different French SSL model sizes (base vs. large), the large architecture yields considerable improvement (nearly 3 to 6 BLEU points) over its base counterpart. When looking into the French SSL models with different amounts of pre-training data (1K, 2.7K, 3K, and 7K), we observe large gains for the base architecture from using 1K to using 2.7K or more pre-training data. There is, however, no significant difference between base models using 2.7K, 3K, and 7K data. Using 7K data even hurts the performance on the pair fr-pt. On the other hand, for the large network, using more data consistently improves the performance on all language pairs. Finally, moving on to task-specific models, Fr-7K-*large* is the best-performing model (or being on par with the best one) in each group. Noticeably, there is a huge improvement when using the ASR fine-tuning approach (c) for the multilingual XLSR-53 model. The method considerably boosts the performance of the multilingual model (compared to using it directly or further pre-training it on the task data) and makes it even on par with the best French SSL models.

## 5.4 Automatic Emotion Recognition (AER)

Automatic Emotion Recognition (AER) research mostly relies on detecting either different emotion categories such as happiness or sadness, or different emotion dimensions such as arousal and valence. Here, we use sequence-to-sequence models on continuous dimensions of emotion.

**Datasets** We use RECOLA [62] and AlloSat [63] datasets as in [9]. RECOLA is a well-known corpus for benchmarking emotion recognition systems, which contains recordings of spontaneous interactions between French-speaking subjects in lab environments. AlloSat is a more recent dataset that contains real-life call center conversations in French. Both datasets are time-continuously annotated by several annotators. The different annotations are averaged to define an emotional dimension *gold-standard*: arousal (from passive to active) and valence (from negative to positive) for RECOLA with a sampling rate of 25 Hz, and a dimensional axis ranging from frustration to satisfaction for AlloSat with a sampling rate of 4 Hz.

**Experiments** In addition to using SSL features, we extracted 40-dimensional MFB features normalized to have zero mean and unit standard deviation over the training set. We used simple regression

Table 7: Concordance Correlation Coefficient of emotion predictions on the RECOLA and AlloSat test sets.

| Features | Corpus - Task | | | | | | | | |
| | RECOLA - Arousal | | | RECOLA - Valence | | | AlloSat - Satisfaction | | |
| | Model | | | | | | | | |
| | LinTh | GRU-32 | GRU-64 | LinTh | GRU-32 | GRU-64 | LinTh | GRU-32 | GRU-64 |
| MFB | .139 | .655 | .649 | .107 | .373 | .421 | .121 | .611 | .612 |
| En-*large* | .465 | .517 | .542 | .154 | .220 | .221 | .102 | .490 | .480 |
| XLSR-53-*large* | .237 | .661 | .669 | .005 | .322 | .200 | .242 | .578 | .582 |
| Fr-1K-*base* | .505 | .654 | .661 | **.243** | .331 | .301 | .403 | .641 | .558 |
| Fr-1K-*large* | .507 | .709 | .708 | .196 | **.555** | .234 | .175 | .601 | .597 |
| Fr-2.7K-*base* | **.521** | **.720** | **.741** | .208 | .498 | **.530** | **.437** | .646 | .687 |
| Fr-3K-*base* | .474 | .700 | .686 | .183 | .388 | .228 | .356 | **.732** | **.740** |
| Fr-3K-*large* | .378 | .267 | .349 | .130 | .202 | .033 | .009 | .468 | .473 |
| Fr-7K-*base* | .502 | .700 | .702 | .214 | .406 | .358 | .394 | .653 | .653 |
| Fr-7K-*large* | .310 | .203 | .078 | .020 | .214 | .068 | .007 | .510 | .474 |

models similar to the ones presented in [9]. The LinTh model only consists of a linear layer followed by a tangent hyperbolic function and the GRU models are 1-layer GRU with the hidden layer $D = [32, 64]$, followed by the LinTh layer. Evaluation metric is Concordance Correlation Coefficient [64] between model predictions and human annotations, as in [65, 66].

**Results** are presented in Table 7. One noticeable result is that, while MFB features cannot reach acceptable performance with the simple LinTanh model, SSL features achieve much better results. As the models get more complex (GRU-32 and GRU-64), the advantage of using SSL features compared to MFB features is less clear. This shows the effectiveness of providing higher level representations (SSL) for AER only when a less complex model (LinTanh) is used. One interesting finding is the ability of the Fr-2.7k-*base* feature to reach close to best results for most cases even though this SSL model has only been trained on non-emotional speech (Fr-2.7K-*base* is trained on a subset of our *medium* set where spontaneous and emotional speech were removed and only read speech was left). Also, since these models are not always better than MFB features when using a more complex model, might show that even though SSL models are able to reach higher level information than MFB, they struggle to extract information related to emotion. We should however highlight the fact that pre-training of 3k models involved less than 1% emotional data (cf. Table 13). Moreover, Fr-1k models, which also only use read speech (but using less data), perform mostly better than Fr-3k and Fr-7k models, which were trained on data containing spontaneous and emotional speech. This shows that by using more data to train SSL models, if mostly non-emotional, we cannot expect better results for the task of emotion recognition. We also observe large variations of performance from one SSL model to another, probably because AER is a very low resource task in this setting. It is thus difficult to conclude on the effectiveness of our SSL models trained on French data compared to the ones trained on multi-lingual or English data. Finally, task-specific pre-training attempts (not reported here) were also made on RECOLA with Fr-3k models but in both self-supervised and ASR based fine-tuning scenarios models did not converge. Further investigations are needed in order to better understand this behavior.

## 6 Discussion

**On societal and environmental impacts.** As an increasing number of NLP papers discussed the potential biases and harms of pre-trained language models and call for more careful design of datasets [67], we set up our large speech corpus with the objective of limiting those in the shared SSL models. First, our speech dataset is carefully documented with relevant metadata (see Table 1) so that it is feasible to analyze the diversity of existing speech sources in terms of social contexts represented (gender, accent, style). As far as gender balance is concerned, we did not manage to have an exact parity in SSL data (our 1k and 3k models have 52% and 38% of female speakers respectively; bigger 7k model do not have enough gender metadata to allow a correct evaluation of gender balance) but we believe the corpus is diverse enough as it was observed that ASR systems, for instance, are overall robust to a certain degree of gender imbalance in the training data [68] (and our gender analysis for ASR confirms this). Also it is worth mentioning that one corpus in our dataset (TCOF) may contain

offensive speech but we believe this is not a problem as we only distribute the SSL models (not the signal). License information is also displayed for *all* sub-corpora (see Table 1). As environmental impact has been highlighted for NLP recently [69], we used for training SSL models the CNRS Jean Zay supercomputer[9] which is a low carbon data center situated in a low carbon area (France). In particular, and following the carbon footprint methodology given in [70], we estimate that 270kg of $CO_2$ was emitted to train our largest 7K model. In comparison, $GPT$-3 may emit 10 Tons of $CO_2$ while being trained in France (*i.e.* lower carbon rate than the USA) [71]. Sharing our seven models mitigates this impact by alleviating multiple training from the community.

**LeBenchmark** We have set up a website[10] for *LeBenchmark* with the aim to: (a) link to the pre-trained models and scripts to reproduce experiments presented in this paper, (b) keep track, through a *Leaderboard*, of future papers and results that would use our evaluation framework, and (c) support contributions for other languages in order to grow *LeBenchmark* dynamically.

**Takeaways** After training our own SSL models for French, we evaluated them on 4 speech tasks (ASR, SLU, AST, and AER). For all of them SSL models were beneficial with respect to conventional filterbank of MFCC features. Tasks such as SLU improved drastically with SSL. We also observed that low and medium resource tasks (SLU and AST) and simpler neural architectures (AER with LinTh) benefited more from task-agnostic SSL features than high resource tasks (ASR). We verified the impact of the language used for pre-training: French SSL models are better than multilingual or English SSL models for ASR, SLU and AST in French. SSL architecture size also matters as *large* models obtained the best performance compared to *base* ones for ASR, SLU and AST. Regarding amount of SSL pre-training data, setting aside AER for which we observe a lot of variability, training on 3k hours is beneficial compared to 1k but jumping further to 7k is less conclusive (*i.e.* improves ASR and AST only, not SLU). As task-agnostic SSL pre-training already provides strong results, we demonstrated that performance can be further improved using task specific pre-training: adding a few iterations of self-supervised pre-training on task specific data allows to improve SLU and AST performance. If transcribed speech is available, it is even better to fine-tune SSL models for ASR on data of interest and then use the obtained model as feature extractor for a downstream task. This worked well for SLU and AST and is, to our knowledge, the first time such a task-specific pre-training is efficiently applied to non-ASR speech systems. Finally, while some SSL models were beneficial to AER, this task needs more exhaustive and reliable evaluations to assess the real impact of SSL.

**Limitations and future work** We currently cover only French language but hope that contributions for other languages would follow in order to grow *LeBenchmark* dynamically. A more fine-grained analysis of the SSL models' performance (beyond single average metric per sub-task) would be also important to fully understand the pros and cons of each SSL model. Finally, as our collection comes with reliable metadata, it should trigger future analysis works on speech SSL such as training gender/style specific models and analyzing speech SSL biases.

# 7   Acknowledgements

This work benefited from the 'Grand Challenge Jean Zay' program and was also partially supported by MIAI@Grenoble-Alpes (ANR-19-P3IA-0003). This paper was also partially funded by the European Commission through the SELMA project under grant number 957017.

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
