# A Appendix

## A.1 Wav2Vec2.0 training behavior

In this Appendix, we report the losses on the development set of the MLS corpus obtained for our different models. Models are stopped at 500.000 steps due to the lack of improvement observed when trained for longer.

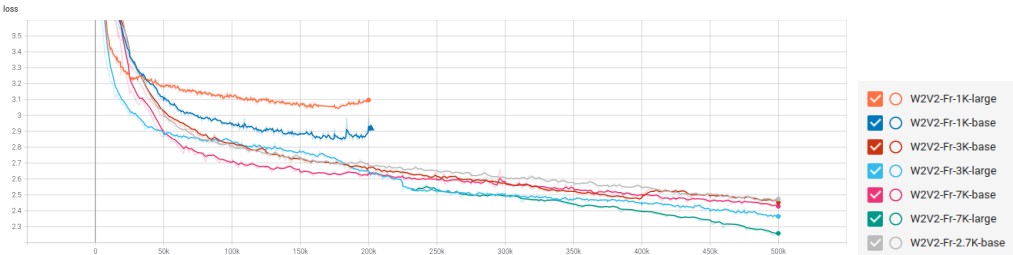

Figure 1: Evolution of the loss on the development set during the pre-training of the SSL models.

## A.2 Architecture and Training details

### A.2.1 ASR: Hybrid DNN-HMM

**Model**

The acoustic models (AM) have been trained trained on 40-dimensional high-resolution (*hires*) MFCC features or SSL feature using the Kaldi toolkit with a state-of-the-art factorized time delay neural network (TDNN-F) architecture [46, 47] on the ETAPE training corpus [44] only. For experiments in this paper, the dimensions of SSL features extracted by *'small'* and *'large'* models are 768 and 1024 respectively. The models have 12 TDNN-F layers (1,024-dimensional, with projection dimension of 128) and a 3,432-dimensional output layer. 100-dimensional speaker i-vectors were appended to the input features for all the models.

**Training Strategy.** The acoustic model was trained using lattice-free maximum mutual information (LF-MMI) [72] and cross-entropy criteria. Speed and volume perturbation have been applied for data augmentation [73]. We used a similar topology to train all systems with different types of input features.

### A.2.2 ASR: End-to-End models

**Model** The end-to-end system fed by 80-dimension log Mel filterbank (MFB) features is based on an encoder/decoder architecture with attention: the encoder is a Convolutional Recurrent Deep Neural Network (CRDNN: VGG + RNN + DNN), and the decoder is a joint CTC/Attention LSTM neural network. For this ASR system, the neural network output corresponds to 500 byte pair encoding (BPE) units [74] computed on the manual transcriptions of the respective training datasets. This model is used to present baseline results from a system that does not use pretrained models.

When used with a pretrained Wav2Vec2.0 model, the end-to-end model is made of this Wav2Vec2.0 model with an additional hidden layer and an output layer on the top. The hidden layer has the same dimension as the Wav2vec2.0 model output (*i.e* 768 for *base* models, 1024 for *large* ones.) The output layer dimension depends on the number of characters in the training data (78 for CommonVoice, 60 for ETAPE).

**Training Strategy** No additional language model is used in these experiments, neither data augmentation. To train (with supervision, by exploiting the manual transcriptions) the Wav2Vec2.0-based end-to-end models, two disjoint Adam optimizers are applied: one to handle the Wav2Vec2.0 pretrained weights and another one to update the randomly initialized weights of the hidden and output layers on the top of the model.

### A.2.3 SLU

**Model** The end-to-end SLU model used in this work is similar to the one proposed in previous works [75, 76, 77, 9]. In particular we use a similar speech encoder employing a pyramidal hierarchy of RNN layers like [78, 9]. The decoder has been also improved integrating two attention mechanisms: one as usual for attending the encoder's hidden states; the other for attending all previous decoder prediction's embeddings, instead of the previous prediction only like in the original LSTM-based encoder-decoder models [58]. Our model is implemented using the *Fairseq* library [42].

**Training Strategy** We use a similar incremental training strategy as [77]. In particular we train first only the encoder of our model for decoding tokens as an ASR model. In order to do so we add a linear layer on top of the encoder which maps the hidden states into the output dictionary size. We use the same dictionary for all symbols in our system. We can thus use the token-level model to initialize parameters of an equivalent model which performs SLU by decoding all together tokens, concepts and their boundaries. For instance, given a sequence of $N$ tokens $w_{j+1}, ..., w_{j+N}$ instantiating a concept $C_i$ in a sentence $S = w_1, ..., w_M$, we use special boundary markers *boc* and *eoc* (for start and end of concept) for each concept, modifying the original sequence into $S = w_1, ..., \text{soc}, w_{j+1}, ..., w_{j+N}, C_i, \text{eoc}, ..., w_M$. This output format has already been used in previous work [79, 80], and it is needed for extracting concept values (or attribute values) together with concepts (or attribute names), as described in [51, 53]. In this work we focus on concept extraction only, we leave the concept value extraction phase for future work. The SLU model trained for predicting the output described above has the same decoder as the token-level model used for initializing its parameters, that is just a linear layer. This first SLU model is used for initializing the parameters of our final SLU model, which has the same encoder, but uses the LSTM decoder described in the previous paragraph. Our training strategy can thus be summarized in the following 3 steps, where the model trained at step $i$ is initialized with parameters of the model trained at step $i - 1$: (1) Encoder+Linear decoder (ASR), (2) Encoder+Linear decoder (SLU), (3) Encoder+LSTM decoder (SLU).

**Implementation details** All models are learned with an Adam optimizer [81], initial learning rate $5e^{-5}$ which is shrinked by a factor of $0.98$ at each training epoch, and batches of size 10 for the first 2 training steps (linear decoder), 5 for the last step (LSTM decoder). Models learn to minimize the CTC loss [59], and we keep the models showing the best error rate on the development data. When learning the final SLU models with a LSTM decoder, we start training with a small warm-up learning rate which is increased linearly up to the initial learning rate during the first 2 epochs. We use this strategy, together with regularization, to avoid *catastrophic forgetting* [82], as these model's encoders are initialized with a model already trained to perform SLU, as mentioned in the previous paragraph. At the decoding phase we average the scores of the 5 best checkpoints on development data.

### A.2.4 AST

**Model** We used a small Transformer [83] architecture having 6 layers of encoder, 3 layers of decoder, and hidden dimension $D = 256$ in all experiments. Following previous work [84, 9], we inserted a block of Linear-ReLU before convolutional layers in the speech encoder for parameter efficiency and model performance reasons.

**Implementation details** Our experiments are performed using the FAIRSEQ S2T toolkit [85]. For text pre-processing, we normalize the punctuation and build 1K unigram vocabularies using Sentencepiece [86] without pre-tokenization. Following common practice [85, 87], utterances having more than 3000 frames are removed for GPU efficiency. All AST models are trained for 500 epochs using the Adam optimizer [88] in which the learning rate is linearly increased for the first 10K warm-up steps then decreased proportionally to the inverse square root of the step counter. The learning rate for all experiments is set to $2 \times 10^{-3}$. We averaged the last 10 checkpoints and used beam search with a beam size of 5 for decoding. The reported results are detokenized case-sensitive BLEU computed using sacreBLEU [89]. As far as task specific pre-training is concerned, for self-supervised fine-tuning (b) we continued training the SSL models on the task data from the last optimizer state for an additional 20K steps. For ASR supervised fine-tuning (c), we used the same hyper-parameters setup as proposed in the original wav2vec 2.0 paper for fine-tuning large models on 100 h of labeled data. We then used the best checkpoints (fine-tuned on the pair fr-en) to extract features, which are the inputs for the downstream AST models.

Table 8: WER results by gender on the ETAPE test dataset for end-to-end ASR.

| Features | WER Male | WER Female | Relative WER difference between Male and Female speakers, % |
|---|---|---|---|
| MFB | 60.2 | 53.6 | 11.6 |
| XLSR-53-*large* | 60.1 | 57.4 | 4.6 |
| En-*large* | 44.3 | 39.3 | 12.0 |
| Fr-3K-*large* | 27.5 | 21.4 | 24.9 |

Table 9: WER results by gender on the ETAPE test dataset for hybrid ASR.

| Features | WER Male | WER Female | Relative WER difference between Male and Female speakers, % |
|---|---|---|---|
| hires MFCC | 32.0 | 20.9 | 21.0 |
| XLSR-53-*large* | 33.3 | 22.6 | 19.1 |
| En-*large* | 30.2 | 19.8 | 20.8 |
| Fr-3K-*large* (task-agnostic pretraining) | 29.7 | 17.7 | 25.3 |
| Fr-3K-*large* (task-specific pretraining) | 26.4 | 15.7 | 25.4 |

### A.2.5 AER

**Implementation details** Training was achieved by Adam optimizer with 250 as the maximum number of epochs; it was stopped after 15 epochs if no improvement over the development set was observed. The loss (and evaluation metric) used here is Concordance Correlation Coefficient [64] between model predictions and human annotations, as in [65, 66]. Sampling frequencies of different features, which was 100 Hz for MFB and 50 Hz for the Wav2Vec models, are different from the sampling frequencies of the annotations. Thus, during the training, we re-sampled the annotations to match the sampling frequency of the features and for testing, we re-sampled the output of the model to match the target annotation. Reported numbers on the paper are averaged results over three different random seeds.

### A.3 Additional Results for ASR

Tables 8 and 9 present the WER by gender reached by different ASR systems on the ETAPE test dataset for end-to-end and hybrid ASR respectively. While in this dataset the WER is lower for female speakers than for male speakers for each ASR system, the relative difference between the results obtained on female voice and the ones obtained on male voices is higher with our Fr-3K-*large* SSL model.

### A.4 Additional Results for SLU

Table 10 reports ASR results on the MEDIA corpus. These ASR models have been used to initialize parameters of basic SLU models with a linear decoder.

Table 11 reports significance test results with the bootstrap method of [49]. As named in the reference paper, the values reported in the table are the *Probability of improvement* (Poi) of a system B over a system A, they can be interpreted as $1-$p-value. In the table system B are "Fr-3K-*large* SV" and "Fr-7K-*large* SV", the two best SLU systems in terms of Concept Error Rate (CER). Between the two best systems, "Fr-7K-*large* SV" seems to be slightly better with a *Poi* of 0.78 over "Fr-3K-*large* SV". But a stronger computation intensive significance test [90, 91] shows that "Fr-7K-*large* SV" and "Fr-3K-*large* SV" are in fact equivalent (p-value of 0.6 in both directions). The same test gave a p-value $< 0.01$ in all the other cases, conforming that "Fr-7K-*large* SV" and "Fr-3K-*large* SV" are indeed the two best models.

Table 10: End-to-end ASR results on the MEDIA corpus (Word Error Rate %).

| Features | Dev | Test |
|---|---|---|
| (from [9]) spectrogram | 35.37 | 35.98 |
| spectrogram | **32.22** | **33.95** |
| **(a) Task agnostic pre-training** | | |
| En-*base* | 19.49 | 20.36 |
| En-*large* | 22.88 | 25.59 |
| Fr-1k-*base* | 21.74 | 23.90 |
| Fr-1k-*large* | 18.01 | 19,29 |
| Fr-2.7k-*base* | 14.23 | 15.40 |
| Fr-3k-*base* | 14.58 | 15.37 |
| Fr-3k-*large* | 11.05 | 11.87 |
| Fr-7k-*base* | 14.18 | 15.22 |
| Fr-7k-*large* | **10.62** | **11.55** |
| XLSR-53-*large* | 15.17 | 16.69 |
| **(b) Task specific pre-training (self-supervised on MEDIA)** | | |
| Fr-3k-*large* | **10.34** | 11.59 |
| Fr-7k-*large* | 10.65 | **11.25** |
| XLSR-53-*large* | 11.71 | 12.58 |
| **(c) Task specific pre-training (fine-tuned for ASR on MEDIA)** | | |
| Fr-3k-*large* | 9.21 | 10.29 |
| Fr-7k-*large* | **9.08** | **9.95** |
| XLSR-53-*large* | 10.63 | 11.45 |

Table 11: Significance tests with the bootstrap method [49] between the two best SLU models, in terms of CER, with respect to all the other models. *SS* and *SV* in the table mean *Self-supervised* (block (b)) and *Supervised* (block (c)) pre-training approaches, respectively.

| **Significance tests (Probability of improvement [49])** | | |
|---|---|---|
| **Tested Model** | Fr-3K-*large* SV | Fr-7K-*large* SV |
| Fr-2.7K-*base* | 1.0 | 1.0 |
| Fr-3K-*base* | 1.0 | 1.0 |
| Fr-3K-*large* | 1.0 | 1.0 |
| Fr-3K-*large* SS | 1.0 | 1.0 |
| Fr-3K-*large* SV | - | 0.78 |
| Fr-7K-*base* | 1.0 | 1.0 |
| Fr-7K-*large* | 1.0 | 1.0 |
| Fr-7K-*large* SS | 1.0 | 1.0 |
| Fr-7K-*large* SV | 0.21 | - |
| XLSR53 | 1.0 | 1.0 |
| XLSR53 SS | 1.0 | 1.0 |
| XLSR53 SV | 1.0 | 1.0 |

# B Details of the corpora used in the paper

Table 12: Corpora/sub-corpora details (at download time). Click on the Corpus name to access its web page.

*t=tokens, w=words, h=hours, min=minutes, sent=sentences, d=dialogues

| Corpus (sub-corpus) name | Identifier (ISLRN, DOI...) | Size* | Modality | Dataset use | License |
|---|---|---|---|---|---|
| African Accented French | SLR57 | 22 h | speech, written | SSL | Apache 2.0 |
| Allosat | | 37 h | speech, written | AER | CC |
| Att-HACK | SLR88 | >300 sent | speech, written | SSL | CC BY-NC-ND |
| CaFE | 10.5281/zenodo.1478765 | 1 h | speech, written | SSL | CC-BY-NC-SA 4.0 |
| CFPP2000 (CEFC complement) | | 20 h | speech, written | SSL | CC BY-NC-SA 3.0 |
| CommonVoice fr_604h_2020-06-22 | | 604 h | speech, written | ASR | CC 0 |
| EPAC | 483-703-007-740-8 | 1677 h | speech, written | SSL | ELRA NC |
| ESLO (ESLO2) | | >400 h | speech, written | SSL | CC BY-NC-SA 4.0 |
| ETAPE | 425-777-374-455-4 | 30 h | speech, written | ASR | ELRA NC |
| GEMEP | | 0.9 h | speech | SSL | academic only, NC |
| MaSS | | ≈ 20 h | speech, written | SSL | MIT License |
| MEDIA | 699-856-029-354-6 | 1,258 d | speech, written | SLU | ELRA NC |
| MLS (French) | | 1,096 h | speech, written | SSL | CC BY 4.0 |
| MPF | | 78 h | speech, written | SSL | CC BY-NC-SA 4.0 |
| mTEDx (fr-*) | SLR100 | 25h - 50h | speech, written | AST | CC BY-NC-ND 4.0 |
| NCCFr | | 35 h | Multimedia, written | SSL | academic only, NC |
| Portmedia (PM_DOM) | 135-793-959-390-8 | 40.5 h | speech, written | SSL | ELRA NC |
| RECOLA | | 9.5 h | Multimedia, written | AER | End User License Agreement |
| TCOF | | 146 h | speech, written | SSL | CC BY-NC-SA |
| Voxpopuli unlabeled | | ≈ 4.5k h | speech | SSL | CC0 |
| Voxpopuli transcribed | | ≈ 215 h | speech, written | SSL | CC0 |

# C   Description of the corpora used in the paper

Table 13: Corpora description.

| Corpus name | Description | Used subcorpus (if existing) |
|---|---|---|
| African Accented French[28] | Recordings of African Accented French speech. | |
| Allosat [63] | The corpus is composed of real-life call center conversations in French and is continuously annotated in frustration and satisfaction. | |
| Att-HACK [29] | This data is acted expressive speech in French, 100 phrases with multiple versions (3 to 5) in four social attitudes : friendly, distant, dominant and seductive. | |
| CaFE [30] | The Canadian French Emotional (CaFE) speech dataset contains six different sentences, pronounced by six male and six female actors, in six basic emotions plus one neutral emotion. The six basic emotions are acted in two different intensities. | |
| CFPP2000 [31] | Interviews in Paris and its suburb. Files not included in the CEFC corpus v2.1, 02/2021. | All CFPP2000 files not in CEFC corpus v2.1, 02/2021 |
| CommonVoice [45] | It is a massively-multilingual collection of read sentences. | French: fr_604h_2020-06-22 |
| EPAC [44] | Conversational speech in French broadcast news. Sub-part from the ESTER Evaluation Campaign (ELRA-E0021). | |
| ESLO [32] | Contains two subcorpora: ESLO1 + ESLO2 (telephone dialogues, public meetings, etc). | ESLO2 |
| ETAPE [33] | Consists of French radio and TV data, selected to include mostly non planned speech and a reasonable proportion of multiple speaker data. | |
| GEMEP [34] | Audio and video recordings featuring 10 actors portraying 18 effective states, with different verbal contents and different modes of expression. | |
| MaSS [39] | The Multilingual corpus of Sentence-aligned Spoken utterances has eight languages, and it is made of audio books from the new testament of the Bible. | French |
| MEDIA [55] | A corpus simulating a vocal tourist information server by a Wizard of Oz system. | |
| MLS [27] | A large multilingual corpus derived from LibriVox audio books. | French |
| MPF [35] | Open corpus, created to study the evolution of French language, the growing of a vernacular language, and the effects of the contacts with immigration languages on French. | |
| mTEDx [60] | The corpus is a collection of audio recordings from TEDx talks in 8 source languages. | fr-en, fr-es, and fr-pt |
| NCCFr [40] | Corpus composed of filmed casual speech conversations between friends. | |
| Portmedia [37] | Human-machine interaction, using the Wizard of Oz technique. Two sub-corpora: PM_LANG: dialogues about tourism in Italian. PM_DOM: dialogues about festival ticket booking in French. | PM_DOM |
| RECOLA [62] | Audio, visual, and physiological recordings of online dyadic interactions between French speaking participants, who were solving a task in collaboration. | |
| TCOF [38] | "Children" sub-corpus : interactions between adults and children (up to 7 years old). "Adults" sub-corpus : interactions between adults. | Adults |
| Voxpopuli [92] | This data was collected from 2009-2020 European Parliament event recordings. | French unlabeled + French transcribed |

# D The Machine Learning Reproducibility Checklist (Ver 1.2, Mar.27 2019)

Table 14: The Machine Learning Reproducibility Checklist, version 1.2
*Possible answers: Yes, No, Not applicable.*

| Status* | To do | Comment |
|---|---|---|
| | **For all models and algorithms presented, check if you include:** | |
| Not applicable | A clear description of the mathematical setting, algorithm, and/or model. | |
| Not applicable | An analysis of the complexity (time, space, sample size) of any algorithm. | |
| Yes | A link to a downloadable source code, with specification of all dependencies, including external libraries. | `https://github.com/ LeBenchmark/NeurIPS2021` |
| | **For any theoretical claim, check if you include:** | |
| Not applicable | A statement of the result. | |
| Not applicable | A clear explanation of any assumptions. | |
| Not applicable | A complete proof of the claim. | |
| | **For all figures and tables that present empirical results, check if you include:** | |
| Not applicable | A complete description of the data collection process, including sample size. | The paper does not report a dataset collection. |
| Yes | A link to a downloadable version of the dataset or simulation environment. | The link to all datasets used in the paper is provided Table 12. |
| Yes | An explanation of any data that were excluded, description of any pre-processing step. | The pre-processing steps are summarized in section 3. |
| Yes | An explanation of how samples were allocated for training / validation / testing. | For all Task, we use the standard partitioning as provided in the datasets. Further information can be found in appendix A.2. |
| Yes | The range of hyper-parameters considered, method to select the best hyper-parameter configuration, and specification of all hyper-parameters used to generate results. | In appendix A.2, the reader can find which hyper-parameters were considered and how their value has beeb chosen. The main reference is the provided github code. |
| Yes | The exact number of evaluations runs. | |
| Yes | A description of how experiments were run. | |
| Yes | A clear definition of the specific measure or statistics used to report results. | Standard, sufficiently well-known, evaluation metrics (WER, CER, BLEU, and CCC) were used. |
| Yes | Clearly defined error bars. | Given that several different tasks were used with different measure, we used different methods to assess the robustness of the results (confidence interval, significance test). These are described in their respective task subsection. |
| Yes | A description of results with central tendency (e.g. mean) and variation (e.g. stddev). | Most result tables provide the global score as well as its variation either in the form of stddev or CI. |
| Yes | A description of the computing infrastructure used. | The supercomputer used is mentioned with an hyperlink providing all the necessary details. |