# OpenReview forum: "Task Agnostic and Task Specific Self-Supervised Learning from Speech with LeBenchmark"
_NeurIPS.cc/2021/Track/Datasets_and_Benchmarks/Round2 — NeurIPS 2021 Datasets and Benchmarks Track (Round 2)_

### Official Review · Reviewer_dULD · 2021-09-18
**Good benchmarks, some errors**

**Rating:** 8
**Confidence:** 4

**Strengths:**

With a little bit more work on the part of the authors to make sure
the data preprocessing is reproducible, this work sems likely to be
the foundation of more progress in self-supervised learning for
speech-related tasks.

**Weaknesses:**

The largest flaw with this paper is that the results do not appear to
be fully reproducible. In particular, the provided github code does
not show how to extract the corpora that make up LeBenchmark. For example, the \
code is commented out here: https://github.com/LeBenchmark/NeurIPS2021/blob/650\
92828dc8e9fcd71d114f6d0a0f125c4b14d4e/ASR/ETAPE-Hybrid/run.sh#L21

Given that LeBenchmark consists of ~20 individual datasets and is
intended as a leaderboard, it looks like a critical error to this
reviewer that scripts to preprocess datasets exactly in the same way
(text normalization, etc.) are not provided in the open source code.


**Additional Feedback:**

Change "41,5" and "16,8" to "41.5" and "16.8" in section 5.2.


**Clarity:**

This is one of the best written papers I've read in a long time.

It is not clear how the LinTh model is comparable to the GRU model,
given that one works on sequence data, and the other works on a fixed
size input. Were the mel filter banks and/or SSL embeddings averaged
before inputting to the linear model? It isn't clear.


**Correctness:**

It is fishy that the ETAPE speech recognition WER results are so high,
but I am not familiar with how challenging this corpus. It would have
been appreciated to see previous papers' results on this corpus for
comparison (even if it wasn't with self-supervised models), to
understand how challenging the task is.


**Documentation:**

I reviewed the hugging face and github repos. Documentation is good enough for reproducibility other than prior concern.


**Ethics:**

No concerns here. The souce data is an amalgamation of existing datasets with licenses clearly described.


**Relation To Prior Work:**

As the authors mention, self-supervised learning for speech is still
very new; there is no existing benchmark (and very little existing
work) in French for this task


**Summary And Contributions:**

This is a benchmark (i.e., a set of tasks with exactly defined data)
for speech recognition, spoken language understanding, speech
translation, and emotion recognition for self-supervised learners. It
provides pre-trained benchmark models, and scripts to recreate them,
but does not release the data used in the benchmark itself or scripts to
preprocess the data into the right format.

---

> ### Author Response · Authors · 2021-09-27
> **Response to Reviewer dULD (our github + high WER on ETAPE + AER question)**
>
> >The largest flaw with this paper is that the results do not appear to be fully reproducible. In particular, the provided github code does not show how to extract the corpora that make up LeBenchmark. For example, the code is commented out here .../…  Given that LeBenchmark consists of ~20 individual datasets and is intended as a leaderboard, it looks like a critical error to this reviewer that scripts to preprocess datasets exactly in the same way (text normalization, etc.) are not provided in the open source code.
>
> We thank the reviewer for this relevant remark. In Table 12 of our paper, we listed all the links for accessing and downloading the data used for training our SSL models. In order to incorporate your suggestion, we updated the repository for our paper  (new directory and README available at https://github.com/LeBenchmark/NeurIPS2021/tree/main/data_preprocessing). There, we included all the scripts we used for preparing all the different datasets. For each dataset, these scripts process the different raw data formats and output a standard json file (there might be additional updates in the coming days on this directory as we are actively working on it).
>
> >It is fishy that the ETAPE speech recognition WER results are so high, but I am not familiar with how challenging this corpus. It would have been appreciated to see previous papers' results on this corpus for comparison (even if it wasn't with self-supervised models), to understand how challenging the task is.
>
> The ETAPE evaluation campaign was the last open French evaluation campaign. During this campaign, the best system got a WER of 21.83% on the test dataset, while the best results presented in our paper are respectively 25.22% for an hybrid HMM/TDNN ASR system and 26.14% for an end-to-end ASR system. But there is an important difference between the data used to train the best ASR system of the official campaign and the one used to train the systems of our paper. While the ETAPE organizers provided 22h of transcribed training data, the participants could use external training data. The winner of the ETAPE ASR evaluation campaign used 511h of speech manually transcribed as training data, to be compared to the 22h used in our paper: we only used the training data distributed by the ETAPE organizers. After your suggestion, we added a comment about comparison to the best submission (+ associated ref) on ETAPE shared task a few years ago.
>
> >"It is not clear how the LinTh model is comparable to the GRU model, given that one works on sequence data, and the other works on a fixed size input. Were the mel filter banks and/or SSL embeddings averaged before inputting to the linear model? It isn't clear."
>
> Since the task is sequence-to-sequence modeling of continuous dimensions of emotion, the inputs and outputs have always the same length (please refer to the Appendix for more technical information on how this is achieved), thus we can use a LinTh model that works across the frames (and not the sequence). Hence the difference between using LinTh model and GRU would only be accounting for the context of the data or not (these two different models were selected purposefully for this difference). We have slightly edited the paper by adding the term "sequence-to-sequence" to clarify that the input and output are both sequences. Thanks for your comment about this.
>
> >Change "41,5" and "16,8" to "41.5" and "16.8" in section 5.2.
>
> We updated the paper accordingly. Thanks.

---

### Official Review · Reviewer_RCTf · 2021-09-20
**A good benchmark**

**Rating:** 7
**Confidence:** 4
**Clarity:** In general, the paper is clear and we…

**Strengths:**

- Large benchmark for speech evaluation

**Weaknesses:**

- The paper is missing the appendix while the main paper refers to it.

**Additional Feedback:**

-

**Correctness:**

- It is not clear how to reproduce the results shown in the paper. The authors need to add important details how to reproduce the results.

**Documentation:**

It is good that the authors provided the link to access the data and code.

**Ethics:**

The paper does not include ethical consideration

**Relation To Prior Work:**

The benchmark focuses on French speech, and it has more variety of datasets.

**Summary And Contributions:**

The paper introduces LeBenchmark, a massive open-source benchmark on eight frameworks on English speech data. The work allows researchers to evaluate speech models on several downstream tasks: Automatic Speech Recognition (ASR), Spoken Language Understanding (SLU), Speech Translation (AST), and Emotion Recognition (AER). In addition, the speech type provided in the benchmark is diverse, making it very useful for evaluating different speech accents and styles.

---

> ### Author Response · Authors · 2021-09-27
> **Response to Reviewer RCTf (missing Appendix and Reproducibility)**
>
> >The paper is missing the appendix while the main paper refers to it.
>
> As required for NeurIPS submissions, the Appendix is available in the Supplementary Material in the form of a .zip file which has been submitted together with the main paper.
>
> >It is not clear how to reproduce the results shown in the paper. The authors need to add important details how to reproduce the results.
>
> Details of each downstream task’s architecture were provided in Appendices A.2.1, A.2.2, A.2.3, A.2.4 and A.2.5 for ASR (hybrid), ASR (end-to-end), SLU, AST and AER respectively. The code is also accessible from http://lebenchmark.com. We also refer the reviewer to the section of our Appendix titled  ‘D The Machine Learning Reproducibility Checklist’.
>
>
> >The paper does not include ethical consideration
>
> We did include ethical considerations in section 6 (paragraph on ‘societal and environmental impacts’).

---

### Official Review · Reviewer_rmCh · 2021-09-21
**A compiled benchmark for evaluating SSL models on French speech data**

**Rating:** 5
**Confidence:** 3
**Correctness:** The evaluation methods and constructi…
**Clarity:** See weakness above.

**Strengths:**

The benchmark created from this paper is a good testbed for French speech models, which is an advanced step towards systematic evaluation of speech models using self-supervised learning.
The framework of the benchmark is well executed with clear scoring scripts, documentations, distribution of pretrained checkpoints, corpora, etc.

**Weaknesses:**

First, the writing could be largely improved w.r.t. the following aspects:
- As the benchmark proposed is specifically designed for French speech data, it would be better to point this out in the title explicitly. Also, I didn't see how this benchmark contributes to multilingual speech models except for French.

- In the experiment sections as well as the supplementals, different tasks employ different neural models. I'd like to confirm that if these models use representations from different pretrained SSL models (i.e. Wav2Vec 2.0) as the inputs and then finetuned on the labeled downstream datasets. It would be more clear if how different models (downstream models v.s. pretrained models) are used for evaluation.
Also, it would be more clear to have a table that compares different hyperparamters of models (architectures / dimensions, etc).

- I would strongly recommend you to replace (braces) with [brackets] for references. Using braces is quite unconventional and hard to read.

Second, the findings and observations in the experiment section are not surprising. I'd like to know why pretraining on more data (7k) is less beneficial for downstream tasks compared to the 3k model, which is a bit counter-intuitive to me. The other question is, why don't you use the same architecture (e.g. transformers) to perform the four tasks but use different models?

**Additional Feedback:**

N/A.

**Documentation:**

Yes.

**Ethics:**

N/A.

**Relation To Prior Work:**

I am not very familiar with speech benchmarks, but it seems that there lacks a section or paragraph that compares with previous speech benchmark datasets, either English or multilingually.

**Summary And Contributions:**

This paper presents a benchmark datasets for evaluating self-supervised learning models on the French speech data. The benchmark is documented with large-scale corpora and a collection of pretrained SSL wav2vec 2.0 models. It includes clear evaluation protocols for four downstream tasks: automatic speech recognition, spoken language understanding, automatic speech translation and automatic emotion recognition.

---

> ### Author Response · Authors · 2021-09-27
> **Response to Reviewer rmCh (several topics)**
>
> >As the benchmark proposed is specifically designed for French speech data, it would be better to point this out in the title explicitly.
>
> We are ambivalent about the suggestion as the target language (French) is already mentioned in the abstract and the name ‘LeBenchmark’ suggests this language as well. Moreover,  our goal is to grow, in the near future, a community around LeBenchmark in order to dynamically augment it with more languages, so we would like to avoid to "set in stone" that this is a French-only benchmark.
>
>
> >Also, I didn't see how this benchmark contributes to multilingual speech models except for French.
>
> LeBenchmark can contribute to the evaluation of multilingual speech models on the downstream tasks it proposes. We did not make another claim related to the contribution to ‘multilingual speech models’.
>
> >In the experiment sections as well as the supplementals, different tasks employ different neural models. I'd like to confirm that if these models use representations from different pretrained SSL models (i.e. Wav2Vec 2.0) as the inputs and then finetuned on the labeled downstream datasets.  It would be more clear if how different models (downstream models v.s. pretrained models) are used for evaluation.
>
> Yes we use representations of the different SSL models as features for the downstream tasks. However those ‘SSL extractors’ are either ‘task agnostic’ (general purpose SSL models) or ‘task specific’ (SSL models fine-tuned on the task data) as explained in the introduction and Section 5. We updated the first paragraph of section 5 to highlight this information. We thank the reviewer for this remark.
>
> >Also, it would be more clear to have a table that compares different hyperparamters of models (architectures / dimensions, etc).
>
> We have added a summary of this information in Table 2 (section 4). Thanks for the suggestion.
>
> >I would strongly recommend you to replace (braces) with [brackets] for references. Using braces is quite unconventional and hard to read.
>
> We thank the reviewer for this suggestion, we updated the paper accordingly.
>
> >Second, the findings and observations in the experiment section are not surprising. I'd like to know why pretraining on more data (7k) is less beneficial for downstream tasks compared to the 3k model, which is a bit counter-intuitive to me.
>
> Setting aside the AER task for which we observe a lot of variability, pretraining on 7k instead of 3k is beneficial for ASR/Hybrid and AST tasks. As far as ASR/end-to-end and SLU are concerned, results on 7k and 3k models are similar. For SLU we show in the Appendix (table 10) that the 7k-model leads to the best results in terms of ASR evaluation on MEDIA. A possible explanation is that we already reached a performance plateau on ASR/end-to-end and SLU with the 3k SSL model, a plateau that the 7k SSL model cannot surpass.
> We leave for future work the investigation of a 7k model trained with more capacity or more updates.
>
> >The other question is, why don't you use the same architecture (e.g. transformers) to perform the four tasks but use different models?
>
> Our goal is to evaluate the real impact of SSL for the best baselines for each task addressed (ASR, SLU, AST, AER). Therefore, for each task, we have a different architecture (described in section 5 & Appendix) which corresponds to the best performance we could obtain using baseline MFCC/FBANK features. A consequence is that we do not have the same architecture (e;g. transformers) for each task. This is in line with SUPERB, an english-only evaluation protocol for SSL models that also uses multiple downstream models. We updated the first paragraph of section 5 to clarify this. Thanks for pointing this out.
>
> >I am not very familiar with speech benchmarks, but it seems that there lacks a section or paragraph that compares with previous speech benchmark datasets, either English or multilingually.
>
> As mentioned in the ‘Background’ section, we are aware of only two similar initiatives for speech SSL models' evaluation: our own preliminary work and the Speech processing Universal PERformance Benchmark (SUPERB) which is contemporary with our work.

---

### Official Review · Reviewer_j8nL · 2021-09-21
**SSL evaluation benchmark for French**

**Rating:** 7
**Confidence:** 3
**Correctness:** Yes.
**Clarity:** Yes, it is well writtened.

**Strengths:**

1. The dataset is quite large and diversified.
2. The evaluation is sufficient and covers multiple domains.
3. This paper finds a good research problem for a fair comparison of different SSL methods.
4. This paper is well written and organized.

**Weaknesses:**

Why did the authors choose French not English for the SSL evaluation? Will it prevent people working on English use your benchmark?

**Additional Feedback:**

N/A

**Documentation:**

Yes, it is well documented.

**Ethics:**

I don't see any ethics issues.

**Relation To Prior Work:**

It is well discussed.

**Summary And Contributions:**

The authors collected a large French speech dataset which consists of a large variety of speech corpora in French that cover different accents, acted emotions, telephone dialogues, read and spontaneous sentences, broadcast speech, and professional speech. It also presents a diversity of problems, information extracted, and annotated resources. It can serve as a good benchmark for SSL tasks.

---

> ### Author Response · Authors · 2021-09-27
> **Response to Reviewer j8nL (Why French?)**
>
> Most existing evaluations of pre-trained SSL models focus on ASR in the English language. In addition to addressing more speech tasks (not only ASR), we chose French for the downstream tasks in order to be able to assess the impact of 3 different SSL models (a) in English (to see if SSL models can transfer to a downstream task in a different language), (b) in multiple languages (to see if multilingual SSL is robust to new languages), and (c) in French (to see if a dedicated SSL model is the best option compared to multilingual case).
>
> The other reason we chose French is that no pre-trained SSL models were available for that language yet, so we believe that this work (which shares 7 pre-trained SSL models in French)  is also an important contribution to speech technology in that language.
>
> Finally, people working on English can use another recently introduced Benchmark (SUPERB) which focuses on English language only (and is a contemporary work to ours).

---

### Author Response · Authors · 2021-09-27
**Paper update**

We thank the 4 reviewers for their useful comments. The pdf of the paper was updated accordingly. We answer below to each reviewer's comment and point, if necessary, to the updates made in the main pdf.

---

### Decision · Program_Chairs · 2021-10-10

**Decision:**

Accept

**Comment:**

All reviewers appreciated the contributions of this work, in particular the diversity of data and domains. Two reviewers do note that more information on hyparams etc. ought to be included, in particular for reproducibility purposes. Several reviewers note that the dataset is in French, and that this may limit adoption of the benchmark; on this point I disagree---we need more datasets in languages other than English (even for similarly high resources European languages). To me, it is a strength of the work that it does not focus on English. This being said, it would be nice if you included a bit more on the unique properties of French (speech) that might be relevant for understanding performance on the benchmark (if there are any). All in all, the work is strong and is likely to be recognized as a good quality SSL speech benchmark.